# Examining New Zealand Unmanned Aircraft Users' Measures for Mitigating Operational Risks

Isaac Levi Henderson 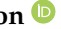

School of Aviation, Massey University, 47 Airport Drive, Palmerston North 4414, New Zealand; i.l.henderson@massey.ac.nz; Tel.: +64-6951-9432

**Abstract:** While the potential risks of unmanned aircraft have received significant attention, there is little in the academic literature that examines how operational risks are mitigated by users. This study examines the prevalence of key operational risk mitigations amongst a sample of 812 unmanned aircraft users in New Zealand, their confidence levels in identifying and complying with airspace requirements, and their ability to read visual navigation charts (VNCs) and use AirShare (a local tool that shows airspace requirements). Significant differences exist between the number and type of mitigations applied, users' confidence levels in identifying and complying with airspace requirements, and users' ability to read VNCs and use AirShare based upon user characteristics. Education, practical assessment, membership of a professional body, professional/semi-professional use, and operating for a certificated organisation all improve risk mitigation (greater number and variety of risk mitigations applied). The only risk mitigation employed by almost all users was conducting a pre-flight check of their aircraft, identifying the need for users to view risk mitigation more holistically. The findings support policy directions related to educational requirements, the ability for member-based organisations and professional bodies to self-regulate, and the fitness of the current regulatory system in New Zealand.

**Keywords:** unmanned aircraft; drones; risk management; airspace requirements; aviation regulation





## 1. Introduction

The potential risks posed by unmanned aircraft operations have received significant attention in the academic literature [1–5], as have the potential benefits of the adoption of unmanned aircraft technologies [6–9]. While there have been studies on how to reduce the risk of unmanned aircraft operations to allow for their benefits to be realised, the current literature has focussed on technical, organisational, and regulatory measures to reduce risk [5,10,11]. However, an important element of risk mitigation is to understand what is already being done by unmanned aircraft users to then assess what measures might be effective for improving risk mitigation. There appears to be a significant gap in the academic literature, whereby user perspectives on risk mitigation are not being addressed, and instead risk mitigation is being studied from the perspective of academic experts and regulators. While these perspectives are highly useful for identifying ideal ways of mitigating risk, one must understand what users are currently doing to be able to recommend how to achieve those ideals. To the author's best knowledge, there is only one study that addresses unmanned aircraft users themselves, examining regulatory compliance in the United States [12]. Accordingly, the purpose of this article is to help ameliorate this gap in the literature by examining how unmanned aircraft users in New Zealand currently mitigate their operational risks. It focusses specifically on eight pre-flight risk mitigations and the use of air band radio as ways that users could reduce ground and airborne risks associated with their operation. While users may mitigate risks in other ways, such as through the use of technological systems (e.g., sense and avoid systems), these mitigations are well-documented in the literature. It also stands to reason that operational

risk mitigations are additive to these and represent the "best practice" even if there are other risk mitigations.

### 1.1. Risks of Unmanned Aircraft Operations

There have been many different perspectives on what risks unmanned aircraft operations might pose, and an equally large literature base on what risk mitigations might be appropriate for mitigating those risks. However, one common way of dividing risks is to think about whether they are airborne risks (i.e., potential for collision with other aircraft) or ground risks (i.e., potential to injure people or damage property on the ground). For the sake of brevity, this article does not provide a detailed discussion of each risk and their associated mitigations, but rather presents Table 1 below, which summarises risks and risk mitigations from selected literature. It is by no means exhaustive, but it does help to illustrate the breadth of risk mitigations that could be considered, and why this article has chosen to focus only on several user-based risk mitigations. It is important to note that Table 1 presents a number of risk mitigations that may seem unrelated at first glance. This is to recognise the holistic nature of an operation, whereby technical (e.g., automatic recovery systems), environmental (e.g., operating in controlled airspace so that air traffic control ensures aircraft separation), organisational (e.g., having procedures and training requirements), and other forms of risk mitigations come together to reduce risk [2,3,13]. Table 1 reflects this holistic viewpoint by presenting all of the risk mitigations together.

**Table 1.** Risk mitigations according to risk type.

| Risk Type | Risk Mitigations | |
|---|---|---|
| Airborne and Ground | Procedures<br>Frangible aircraft<br>Flight planning<br>Pilot situational awareness<br>Maintenance<br>Organisational procedures<br>Night operations<br>Automated recovery systems<br>Failure warning systems<br>Familiarisation with the operating area<br>Geo-fencing<br>Cyber security<br>Protected area | User operational planning<br>High visibility paint<br>Maintenance check of aircraft<br>Maintenance check of equipment<br>Pilot training<br>Airworthiness<br>Aircraft size<br>Aircraft weight<br>Aircraft velocity<br>Aircraft type<br>Emergency systems<br>Low altitude operations |
| Airborne | Airspace segregation<br>Air traffic services<br>Visual separation<br>Collision avoidance system<br>Radio broadcasts<br>ADS-B In and Out<br>Navigation lighting<br>Listening to air band radio<br>Visual observers | Issue a NOTAM<br>Read NOTAMs<br>Operate in restricted airspace<br>Operate in danger areas<br>Operate in controlled airspace<br>Engagement with local airspace users<br>Operate where there are low activity levels<br>Chase plane |
| Ground | Operate away from people and property<br>Establish recovery or ditching points<br>Containment systems<br>Surveying operating environment<br>Notifying public<br>Parachute systems | Personal protective equipment<br>Emergency response equipment<br>First aid training<br>Low population density<br>Glide capability |

Abbreviations in table: ADS-B stands for Automatic Dependent Surveillance-Broadcast, NOTAM stands for Notice to Airmen. Sources: [1,2,4,7,10,14–23].

## 1.2. New Zealand as a Study Location

Prior to outlining the operational risk mitigations under study, it is important to understand the context for where the study was conducted. New Zealand provides an interesting case study because its regulations are broadly consistent with the model regulations for unmanned aircraft proposed by the International Civil Aviation Organisation (ICAO) [24]. The Civil Aviation Authority of New Zealand (CAANZ) uses Civil Aviation Rule (CAR) Part 101 to provide general operating rules for unmanned aircraft users [25]. CAR101.13 specifically requires persons operating unmanned aircraft to "take all practicable steps to minimise hazards to persons, property and other aircraft" [25] (p. 11). For users that cannot comply with the Part 101 rules, CAR Part 102 provides a means of becoming a certificated organisation through the submission of an exposition and accompanying documents that outline how the organisation is managed and intends on operating and maintaining its aircraft, among other things [26]. However, the requirement to minimise hazards is not a rule that can be varied. There are a few differences with ICAO's model regulations, such as there being no registration, no differentiation between recreational and commercial operations, and no use of Part 149 (regarding member-based organisations) to regulate model flying clubs (instead this is done using grandfathered rights under CAR Part 101). Because of the relatively small number of differences, the regulatory environment of New Zealand makes for a suitable case study location as the findings may be broadly applicable to other countries that model their regulations from ICAO. While there are still discrepancies around regulations (due to the lack of international standards), there are nonetheless commonalities in what are identified as suitable risk mitigations for different operations and the requirement to mitigate risks as far as reasonably practicable. For example, the rules applicable in the United States, European Union, and Canada, as well as international bodies such as the Joint Authorities for Rulemaking of Unmanned Systems (JARUS), all reflect a similar approach to identifying and mitigating risks, even though the specific rules and mitigations vary [27–30]. Thus, while it will not be possible to directly translate from New Zealand to other countries, the results are likely to be analogous.

## 1.3. User-Based Operational Risk Mitigations

As noted earlier, this article focusses on eight pre-flight mitigations as well as air band radio. These are all risk mitigations that are user-based. In other words, they do not stem from the aircraft technology being used nor the regulatory environment; rather, they are actions that users might take to mitigate risks associated with their operation. This article does not argue that some risk mitigations are better than others for users to employ (due to the differing operations conducted) and instead argues that the number of risk mitigations employed is a more important metric because each particular risk mitigation addresses different types of risks. This argument parallels those made in other safety-sensitive operating environments, such as construction, maritime, and research laboratories [31–33]. Below, each risk mitigation included in this study is described and its inclusion justified.

### 1.3.1. AirShare

AirShare provides a number of services related to unmanned aircraft operations in New Zealand, including a repository of commercial operators, advice on regulations, and other services [34]. However, the relevant services in terms of risk mitigations are the provision of a New Zealand airspace map on their website and phone app, allowing flights to be logged using their service (including requesting permission to fly in controlled airspace), and ability to access NOTAMs and other flight advisories [35–37]. Participants who log their flights on AirShare are thus mitigating risk by ensuring that they check the airspace they intend to fly in and any pertinent information about that airspace. This arguably increases situational awareness and their ability to assess risk.

### 1.3.2. Visual Navigation Charts (VNCs)

VNCs provide a visual depiction of aeronautical information, including the boundaries of controlled airspace, locations of published aerodromes, locations for different activities (e.g., parachute landing areas), special use airspace (e.g., low fly zones, military operating areas), and other pertinent information [38,39]. Pilots of unmanned aircraft can use VNCs to plan their operations according to airspace requirements and to enhance situational awareness by being knowledgeable about other potential operations that may occur in the area. These aspects of flight planning and familiarisation with the operating area have previously been identified as effective risk mitigations [1]. VNCs differ from AirShare as a form of airspace information because they are far more comprehensive (also being used by manned aircraft pilots) and because they are simply a map, whereas AirShare also allows for the logging of flights and the ability to obtain permission to fly in controlled airspace.

### 1.3.3. Notices to Airmen (NOTAMs)

NOTAMs are notices that concern the "establishment, condition or change in any aeronautical facility, service, procedure or hazard, the timely knowledge of which is essential to personnel concerned with flight operations" [40] (p. 62). Reading NOTAMs has been identified as a key safety barrier to help prevent mid-air collisions between manned and unmanned aircraft [2]. In New Zealand, NOTAMs are issued when unmanned aircraft fly higher than 400 feet above ground level in Class G airspace under Part 101 and are also often issued by organisations certificated to undertake more specialised operations under Part 102 [25,26]. This is done to make other airspace users aware of such operations. Unmanned aircraft users should check NOTAMs prior to flights to ensure that there are no changes to airspace requirements and that they are aware of any temporary hazards that may affect the safety of their operations (or vice versa). Unlike VNCs and AirShare, NOTAMs concern temporary changes to airspace information and are updated daily.

### 1.3.4. Job Safety Assessments (JSAs)

One of the requirements for operating under CAR Parts 101 or 102 is to take all practicable steps to minimise hazards to persons, property, and other aircraft [25,26]. JSAs are used as a formal way of identifying risks associated with a particular operating area and ensuring mitigations are in place. Figure 1 shows an example of what one might look like, noting that they vary markedly based upon the nature of the operation.

JOB SAFETY ASSESSMENT

| Company | | | Date | |
|---|---|---|---|---|
| Task | Location | | Check the following and address as needed | |
| | | | | Check as Completed ✓ |
| | ✓Sketch of area (if necessary) | | Maps and charts available and checked | |
| | | | Weather, within limits for machine and operation | |
| | | | NOTAMs checked | |
| | | | NOTAM required?  Issued Y/N | |
| | | | Possibility of public moving into area | |
| | | | Footpath/right of way | |
| | | | Landing area including alternate | |
| | | | Ability to maintain 30m of public | |
| | | | Obstructions (buildings Trees) | |
| | | | Possible interference (powerlines/antennae) | |
| | | | Ability to maintain visual line of sight | |
| | | | Controller's ability matches location/task | |
| | | | Permission of any landowners | |
| | | | Privacy | |
| | | | Local restrictions, by laws | |
| | | | Need for signage | |
| | | | *Any additional requirements* | |
| Pilot | Signature | | | |
| Crew | | | | |
| Comments: | | | | |

**Figure 1.** Example of a JSA [41].

### 1.3.5. Pre-Flight Checks

Pre-flight checks are an important part of ensuring that an unmanned aircraft is airworthy prior to flight, with unmanned aircraft pilots usually undertaking pre-flight checks in accordance with the user guide for the aircraft in question [6]. Failure to perform (or correctly perform) pre-flight checks has been linked to greater likelihood of aircraft accidents [42,43], with anecdotal examples showing similar potential for unmanned aircraft accidents [44–46]. Thus, pre-flight checks are a critical risk mitigation for ensuring an aircraft is airworthy prior to flight.

### 1.3.6. Model Flying New Zealand (MFNZ) Site-Specific Requirements

Model aircraft hobbyists often join model flying clubs, which have existed for many decades. In New Zealand, MFNZ is the largest national body for model aircraft hobbyists. Members must abide by the MFNZ members manual, national safety code, and other internal procedures [47,48]. Due to their long history, model aircraft organisations have been allowed to self-regulate in many countries [49], with New Zealand being no different in this regard (although CAANZ does maintain regulatory oversight, there are just special permissions for MFNZ). In this sense, the internal procedures become a risk mitigation for MFNZ members, with the internal procedures covering where members can operate, how they can operate, and what levels of training are required for different types of operations. When browsing their members manual [47], one can see many of the risk mitigations identified in Table 1 being applied. Often operations are done in danger areas that are designated for model aircraft use and shown on VNCs for all pilots to see. Pilots of aircraft (both manned and unmanned) are advised to avoid danger areas when possible, and to carefully consider the hazards present if they must enter the area [38]. The operation of unmanned aircraft within danger areas has previously been identified as an effective way of reducing the risk of mid-air collisions by ensuring that pilots are aware of unmanned aircraft operations as a potential hazard in that area [2]. However, MFNZ site-specific requirements do go beyond relying solely on this risk mitigation.

### 1.3.7. Internal Company Procedures

Unmanned aircraft are being used in many different professions for varying applications, such as in forestry, construction, medicine, photography, and agriculture [7,9,39,50,51]. In many of these instances, the operator of the unmanned aircraft will not be operating under the authority of a Part 102 Operator's Certificate because the operation can be done within the Part 101 rules (which anyone can operate under). However, many companies decide to go beyond the Part 101 rules and outline internal procedures for operating an unmanned aircraft as an employee of the company. Usually, such procedures are created to protect the organisation against reputational damage (e.g., a high-profile accident), ensure employees and the public are safe, to keep insurance premiums down, and to ensure compliance with other health and safety legislation. Such procedures vary widely, but often encompass training requirements to ensure sound knowledge of the Part 101 rules, and operational procedures that ensure risk is at an acceptable level.

### 1.3.8. Part 102 Procedures

Organisations that wish to conduct operations outside of the Part 101 rules may wish to apply to become certificated under Part 102 [26]. This process involves submitting a document called an exposition, which outlines how the organisation is managed, how it intends on operating unmanned aircraft in different sorts of airspace and operating areas, training standards, maintenance standards, and a safety management system [6]. There is a model exposition provided by CAANZ [41]. The combination of procedures brings risks down to an acceptable level to be able to perform operations that are not permitted for the general public. For example, to be able to operate above people without consent, a combination of risk mitigations such as training, maintenance, parachutes, automatic

recovery systems, or others can be used to show that an equivalent level of safety can be achieved as compliance with the Part 101 rules.

### 1.3.9. Air Band Radio

Use of air band radio is often considered second only to visual lookout in aviation [52]. Listening to air band radio allows for pilots to build a mental picture of what traffic is in the area, which then assists in collision avoidance and visually detecting other aircraft [53]. However, it is not free of issues with many aviation accidents featuring miscommunication as a contributing factor [54]. Nonetheless, it is a tool that can help enhance situational awareness for unmanned aircraft users when they are operating in areas where manned aircraft may be operating [2], such as within 4 km of an aerodrome, higher than 400 ft above ground level, and within controlled airspace [6].

### 1.3.10. Summary of User-Based Operational Risk Mitigations

Table 2 summarises the operational risk mitigations that are investigated as part of this study in terms of their potential to mitigate airborne and ground risk. A tick indicates that it reduces that form of risk, while a hyphen indicates no effect.

**Table 2.** Summary of operational risk mitigations.

| Risk Mitigation | Reduces Airborne Risk | Reduces Ground Risk |
|---|:---:|:---:|
| Logging flight on AirShare | ✓ | - |
| Checking VNCs | ✓ | - |
| Checking NOTAMs | ✓ | - |
| Conducting a JSA | ✓ | ✓ |
| Pre-flight check | ✓ | ✓ |
| MFNZ internal procedures | ✓ | ✓ |
| Internal company procedures | ✓ | ✓ |
| Part 102 procedures | ✓ | ✓ |
| Air band radio | ✓ | - |

### 1.4. User Characteristics

Because this study focusses on users' measures for mitigating operational risks, it is also important to understand differences that might exist between different types of users. This section details some of the ways that participants are grouped in this study and why those groupings are important.

### 1.4.1. User Type

Users of unmanned aircraft may be operating them for different purposes. Participants were asked to identify whether they were recreational users, semi-professional users (where less than 50% of work time is spent on activities related to an unmanned aircraft), or professional users (where more than 50% of work time is spent on activities related to an unmanned aircraft). This is because these three types of users are likely undertaking operations with different risk profiles, and thus may be employing different risk mitigations. Note that the differentiation between semi-professional and professional was also based upon time rather than income, as some operations may be not-for-profit, such as scientific research or humanitarian operations. This study finds that semi-professional and professional users will use a greater variety of risk mitigations as well as a higher overall number of risk mitigations.

### 1.4.2. Courses

There are several providers of courses on unmanned aircraft within New Zealand, with courses aimed at providing general aviation knowledge and knowledge specific to unmanned aircraft operations [55]. This study illustrates that those who have attended a course are more likely to apply a range of risk mitigations, as well as a higher overall

number. This is because a better understanding of unmanned aircraft operations should provide a better understanding of how to mitigate associated risks.

### 1.4.3. Operational Competency Assessments (OCAs)

OCAs measure practical competency at flying an unmanned aircraft in a similar way to flight examinations for manned aircraft pilots. While primarily aimed at flying ability, soft skills like risk assessment and risk management are also assessed prior to and during the assessment. Accordingly, this study demonstrates that those who have completed OCAs use a greater variety and a higher number of risk mitigations.

### 1.4.4. MFNZ Members

Members of MFNZ are bound by their members manual and other internal procedures. These existed prior to the wider regulation of unmanned aircraft due to their perceived ability to self-regulate. Consequently, this study finds that MFNZ members are associated with using MFNZ site-specific requirements as a pre-flight risk mitigation and are less likely to use other risk mitigations.

### 1.4.5. UAVNZ Members

UAVNZ is an industry and professional body representing the commercial unmanned aerospace sector within New Zealand [56]. Its members comprise operators, manufacturers, researchers, and providers of support services. Their members receive no special consideration from the regulator, although they do make submissions on government policy to advocate on behalf of the sector. However, as a professional body, UAVNZ has a code of conduct that all new members must sign before their membership can be considered [57]. While it is not as prescriptive as MFNZ's members manual, it does place a strong emphasis upon safety and risk mitigation. Thus, there may be differences in the number and variety of risk mitigations employed by UAVNZ members because their membership could be revoked if they fail to abide by the code of conduct.

### 1.4.6. Part 102 Certification

Those who operate under the authority of a Part 102 Operator's Certificated do so in line with their organisation's exposition, which outlines specific risk mitigations for different types of operations. Because these users will be conducting operations that fall outside of the Part 101 rules and have taken a risk-based approach to ensure that an equivalent level of safety is achieved, it is likely that they will employ a greater number and variety of risk mitigations compared with other users.

## 2. Materials and Methods

This article forms a part of a wider piece of research investigating unmanned aircraft users within New Zealand. The data were obtained from a large online survey of unmanned aircraft users in New Zealand, which asked a variety of questions regarding how unmanned aircraft are used, ownership rates, user characteristics, risk mitigations, and opinions on the regulatory system. The survey ran from October 2020 until January 2021 and yielded 919 results, with the raw data made publicly available [58]. The full list of questions can also be found at this link: (https://doi.org/10.6084/m9.figshare.16571558.v1, accessed on 12 January 2022); however, Appendix A provides the list of questions that are used as the basis for this article. To be recruited to participate in the survey, the following criteria had to be met:

1. Must be resident in New Zealand;
2. Must have flown an unmanned aircraft before;
3. Must be 16 years or older.

In addition to the recruitment criteria for the overall research project, this specific study had the following exclusion criteria:

1.　Participants had answer until at least question 9 in Appendix A, otherwise their response would not be meaningful for this study;
2.　Participants had to be current users of an unmanned aircraft (i.e., excluding past users).

After applying the exclusion criteria for this study, 812 participants' data were extracted from the dataset. Of these participants, 795 (97.91%) were male, 12 (1.48%) were female, 1 (0.12%) identified as transgender, and 4 (0.49%) preferred not to say. The mean age was 57.02 years (*SD* = 15.46), with the youngest participant being 16 and the eldest 88. Recreational users formed most of the sample with 716 participants (88.18%), while 53 (6.53%) identified as semi-professional (less than 50% of work time spent on unmanned aircraft operations), and 43 (5.30%) identified as professional (more than 50% of work time spent on unmanned aircraft operations). In terms of levels of training, 391 participants (48.15%) had completed a theory course on unmanned aircraft operations, and 579 participants (71.31%) had completed an operational competency assessment or flight examination. MFNZ members made up a majority of the sample with 650 participants (80.05%), while a minority of 115 participants (14.16%) were members of UAVNZ. Only 95 participants (11.70%) had operated under the authority of a Part 102 Operator's Certificate before. Because the groups (aside from user type) are not mutually exclusive, a participant may fall within multiple groupings. Table 3 presents a cross-tabulation of these demographic variables to see how user characteristics overlap.

**Table 3.** Cross-tabulation of demographic variables.

| Demographic Variable | N | Course | OCA | MFNZ | UAVNZ | Part 102 |
|---|---|---|---|---|---|---|
| Recreational User | 716 | 313 (43.72%) | 501 (69.97%) | 61 (86.82%) | 88 (12.29%) | 53 (7.40%) |
| Semi-Professional User | 53 | 37 (69.81%) | 38 (71.69%) | 16 (30.19%) | 13 (24.53%) | 10 (18.87%) |
| Professional User | 43 | 41 (95.35%) | 40 (93.02%) | 15 (34.88%) | 14 (32.56%) | 32 (74.42%) |
| Completed Theory Course | 391 | - | 349 (89.26%) | 321 (82.10%) | 74 (18.93%) | 74 (18.93%) |
| Passed OCA | 579 | 349 (60.28%) | - | 516 (89.12%) | 86 (14.85%) | 91 (15.72%) |
| MFNZ Member | 650 | 321 (49.38%) | 516 (79.38%) | - | 98 (15.08%) | 68 (10.46%) |
| UAVNZ Member | 115 | 74 (64.35%) | 86 (74.78%) | 98 (85.22%) | - | 32 (27.83%) |
| Part 102 Operator | 95 | 74 (77.89%) | 91 (95.79%) | 68 (71.58%) | 32 (33.68%) | - |

Note: percentages were calculated by dividing the number in each column by the N for the corresponding row.

Because the different questions in Appendix A provide different types of data, different statistical analyses were performed based upon the question type. Questions 3–8 in Appendix A were used to categorise participants into groups based upon different user characteristics. Questions 9–11 were treated as categorical and participants were coded as either "yes" or "no" for whether they typically employ a particular risk mitigation. Chi-squared tests of independence were used to see whether statistically significant associations exist between demographic groups and the use of particular risk mitigations, with effect size being reported with Cramer's V [59]. The number of pre-flight risk mitigations (i.e., all except for air band radio) was also recorded as a sum of the number of the "yes" categorisations for the options covered in question 9, with 8 being the maximum score. This is because each risk mitigation addresses slightly different risks, so one can argue that

the greater the number typically applied, the higher one's risk mitigation, following similar arguments in other safety-sensitive environments [31–33].

Question 12 uses a Likert scale, and each option was coded from 1 to 5 from least to most confident at being able to identify and comply with airspace requirements. This was treated as both ordinal and continuous due to conflicting opinions in the literature [60,61], with the use of a Chi-squared goodness-of-fit test [62] and a single-sample *t*-test [63].

Questions 13 and 16 ask participants whether they can read a VNC or use AirShare, and questions 14 and 15 for VNCs and 17 and 18 for AirShare assess whether the participant can correctly read excerpts from each source of airspace information (see Figures A1–A4 in Appendix A). Correct answers to each question are indicated with "**[correct answer]**" next to the correct option in Appendix A. For both VNCs and AirShare, participants were coded as either getting both questions correct, getting one correct, or getting both wrong. Answering "unsure" to any of these questions was treated as incorrect. Thus, VNC and AirShare accuracy can be used as ordinal measures.

Kruskal–Wallis H tests [64] were used to see whether differences existed between participants' number of typical risk mitigations, levels of confidence in their ability to identify airspace requirements, levels of VNC accuracy, and levels of AirShare accuracy based upon their user type. Distributions were checked for similarity by visual inspection of a boxplot. Pairwise comparisons were performed using Dunn's procedure [65] with a Bonferroni correction [66] for multiple comparisons.

Mann–Whitney U tests [67] were performed to see whether differences existed between participants' number of typical risk mitigations, levels of confidence in their ability to identify airspace requirements, levels of VNC accuracy, and levels of AirShare accuracy based upon whether they had attended a course, completed an OCA, were members of MFNZ or UAVNZ, or had operated under a Part 102 Operator's Certificate before. Distributions were assessed to be similar based upon visual inspection. Results are reported according to mean ranks and distributions using an exact sampling distribution for *U* [68].

To examine the relationship between accuracy in interpreting VNCs and AirShare and confidence levels, cumulative odds ordinal logistic regressions [69] were run with VNC/AirShare accuracy scores as the dependent variable and confidence scores as the independent variable.

## 3. Results

### 3.1. Typical Pre-Flight Operational Risk Mitigations

All participants typically applied at least one risk mitigation prior to an operation; however, there were still differences between the number of participants applying each risk mitigation. Figure 2 presents the prevalence of each risk mitigation being typically applied by a user, where "yes" indicates they do typically use it and "no" indicates that they do not typically use it. The Chi-squared tests of independence, where significant, are also presented in terms of their directionality. An up arrow and green text means that a user group is more likely to employ that risk mitigation, while a down arrow and brown text implies that they are less likely to do so. Appendix B.1 presents additional information related to these Chi-squared tests of independence.

Figure 3 presents the statistically significant results of the Kruskal–Wallis H test with Dunn's pairwise comparisons, as well as the statistically significant results of Mann–Whitney U tests regarding differences in the number of risk mitigations typically applied according to user characteristics. Full statistical information is provided in Appendix B.2.

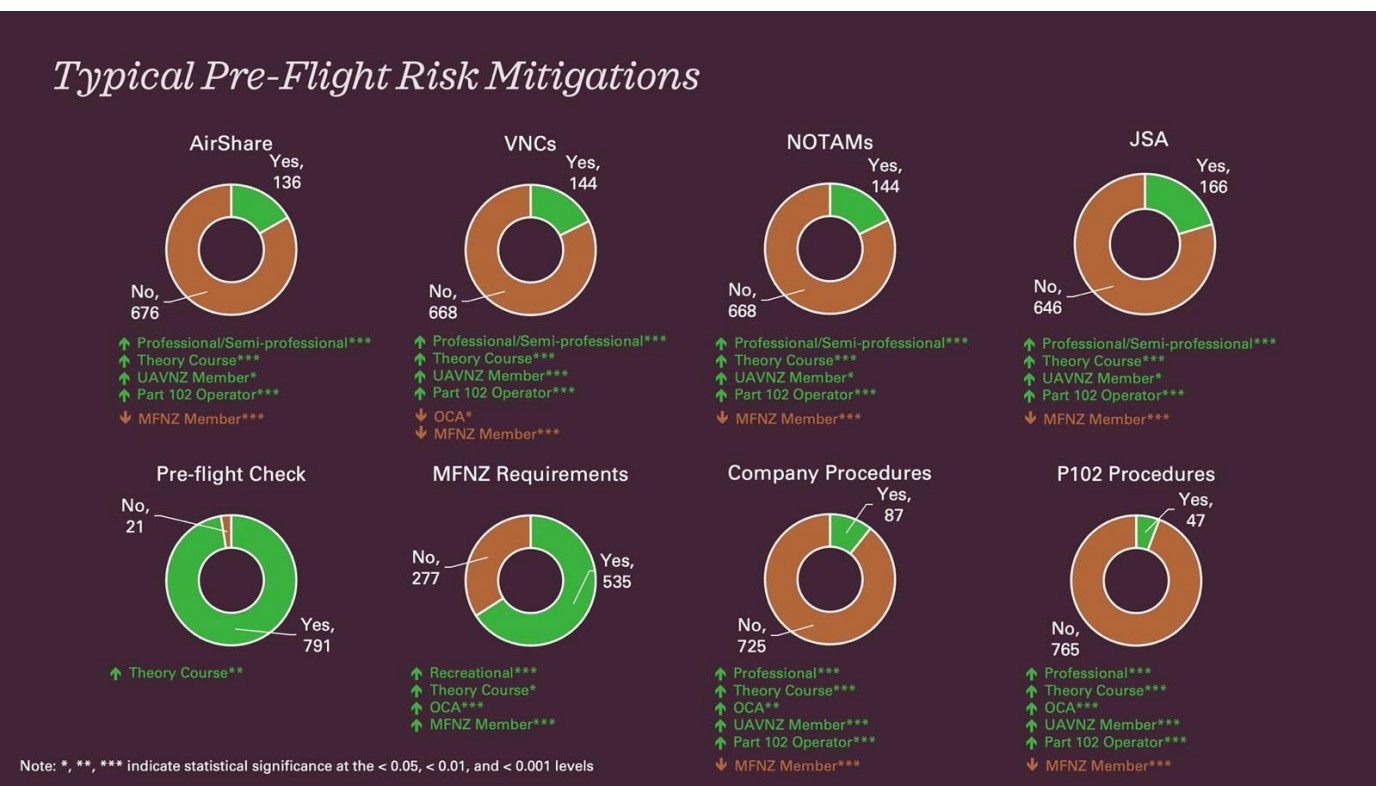

**Figure 2.** Users' typical pre-flight risk mitigations along with groups associated with being more or less likely to use each pre-flight risk mitigation.

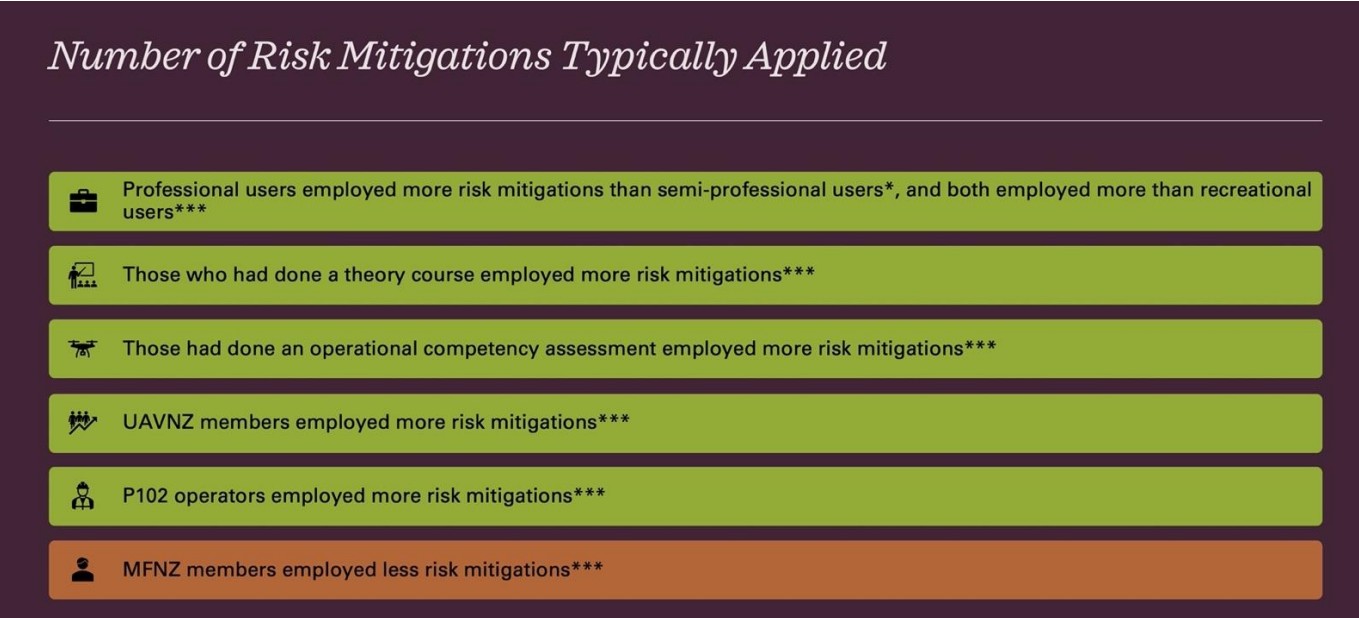

**Figure 3.** Differences in the number of risk mitigations typically applied based upon user characteristics. Note: *, *** indicate statistical significance at the <0.05, <0.0001.

### 3.2. Use of Air Band Radio

There were 210 participants (25.86%) who indicated that they use air band radio for some operations. However, only 207 participants answered the follow-up question to see which types of operations they typically use air band radios. Chi-squared tests of independence revealed significant associations between the use of air band radio as well as the scenarios for which it is typically used and user characteristics. The descriptive

data as well as the results of the Chi-squared tests of independence are shown in Figure 4. Additional statistical information associated with the Chi-squared tests of independence is reported in Appendix B.3.

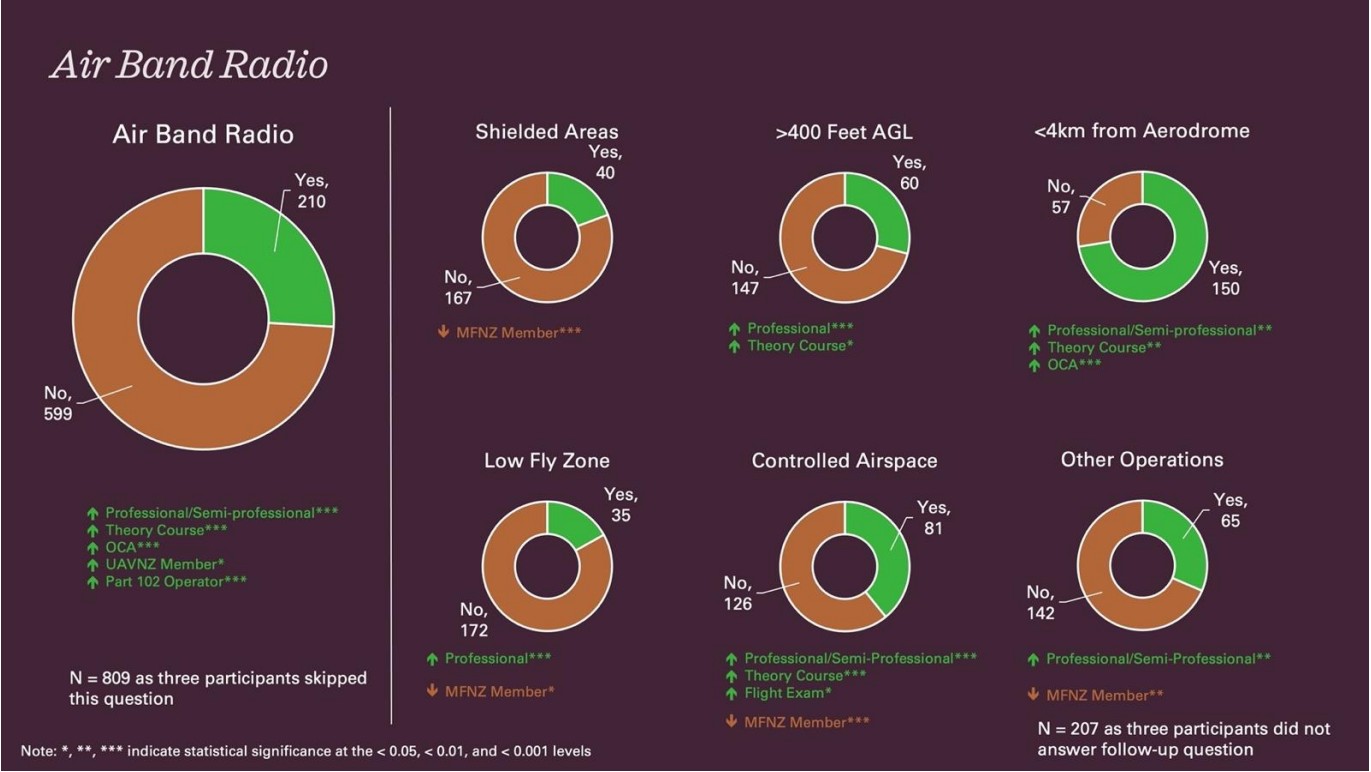

**Figure 4.** Use of air band radio in general and in specific scenarios.

### 3.3. Confidence in Identifying and Complying with Airspace Requirements

There were 808 participants who answered the Likert scale question regarding their confidence levels in their ability to identify different types of airspace and comply with their requirements. The single-sample *t*-tests and Chi-squared goodness-of-fit tests revealed that the mean and mode were statistically significantly different from the null hypothesis, with confidence levels being relatively high. Kruskal–Wallis H tests and Mann–Whitney U tests also revealed differences in confidence levels based upon user characteristics. Abbreviated results are reported in Figure 5, where the up arrow means the group had higher confidence levels, and the hyphen indicates no difference from the rest of the sample. Full statistical results are reported in Appendix B.4.

### 3.4. Ability to Read VNCs and Use Airshare

There were 804 participants who answered whether they know how to read VNCs, and 761 who answered whether they ever use AirShare to check airspace requirements. In both instances, those who said that they could were asked follow-up questions to verify their accuracy (see questions 14 and 15 for the VNC excerpts, and questions 17 and 18 for the AirShare excerpts in Appendix A that were used to assess accuracy); however, there was a drop off in the number of participants answering the follow-up questions. Only 282 of the 325 that said they could read VNCs and 213 out of the 223 participants who said they could use AirShare answered the follow-up questions. Whether this drop-off was due to lack of ability or simply the inconvenience of having to answer more questions is unknown. Figure 6 presents the accuracy of participants according to the number of follow-up questions they answered correctly. Statistically significant results from the Kruskal–Wallis H tests, Mann–Whitney U tests, and cumulative odds ordinal logistic

regressions are also presented in Figure 6, with full statistical reporting in Appendix B.5. (VNCs) and Appendix B.6. (AirShare).

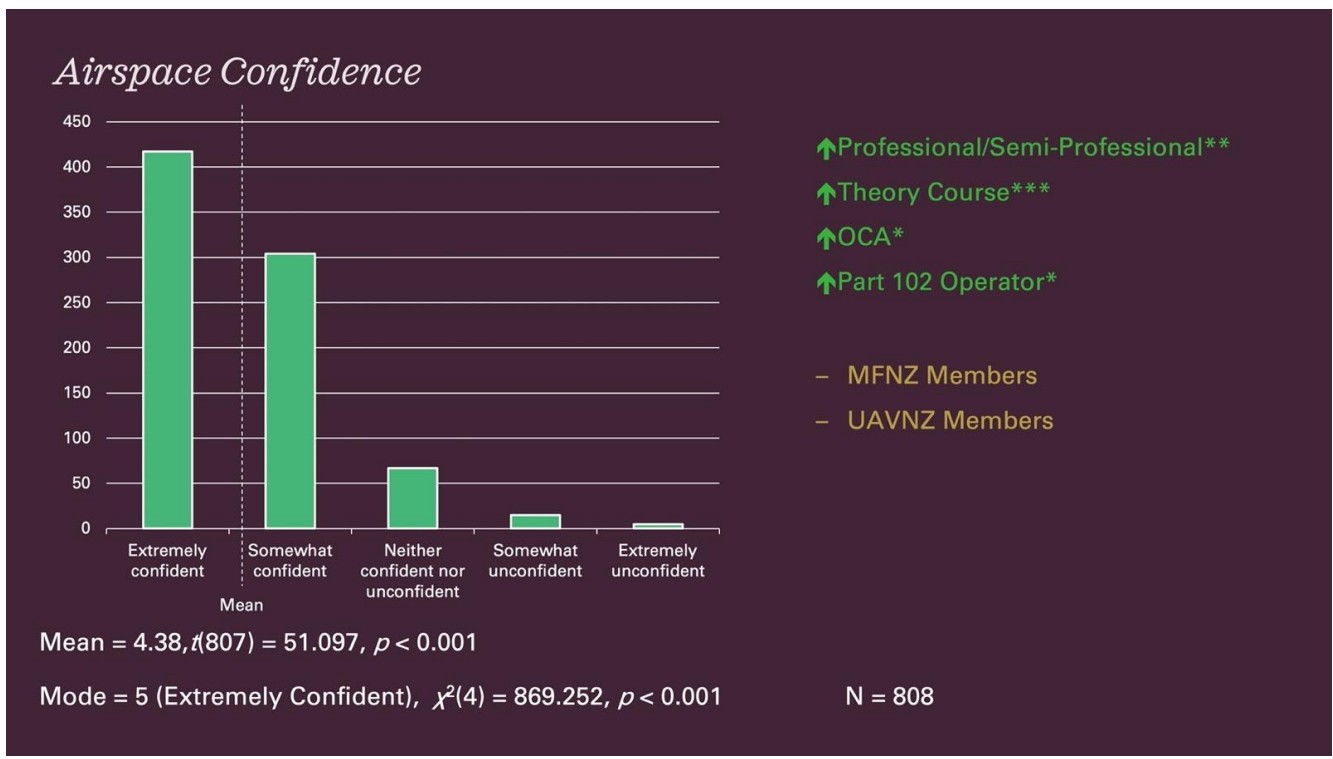

**Figure 5.** Users' confidence in being able to identify and comply with airspace requirements. Note: *, **, *** indicate statistical significance at the <0.05, <0.01, <0.0001.

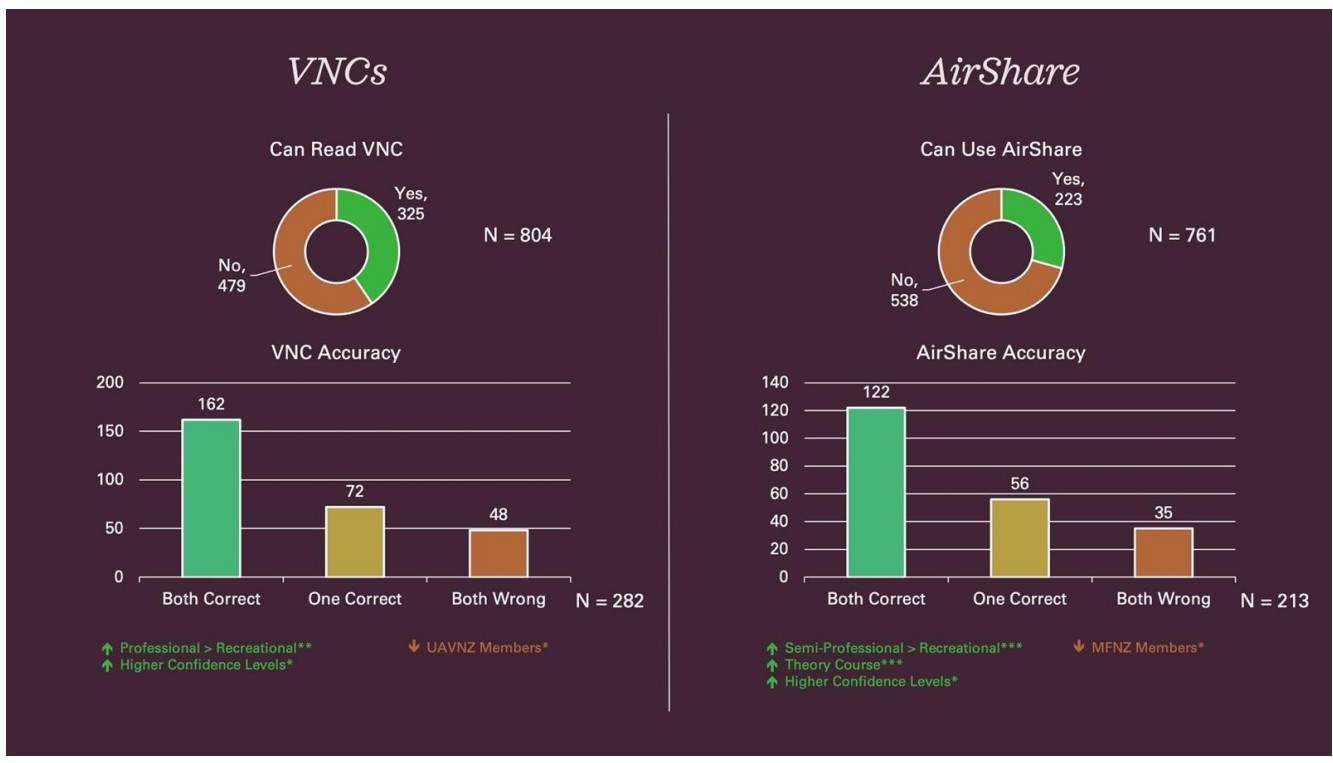

**Figure 6.** Users' ability to read VNCs and use AirShare. Note: *, **, *** indicate statistical significance at the <0.05, <0.01, <0.0001.

## 4. Discussion

### 4.1. Are Pre-Flight Checks Enough?

This study finds that the most common risk mitigation was pre-flight checks, which are typically undertaken by 97.41% of users prior to an operation. Pre-flight checks are certainly a useful risk mitigation; however, the relatively low number of users that typically use AirShare (16.75%), check VNCs (17.73%), check NOTAMs (17.73%), conduct JSAs (20.44%), and use air band radio (25.86%) indicates that perhaps too much reliance is being placed upon pre-flight checks and users need to think about risk mitigation more holistically. However, it could simply be that most users do not operate unmanned aircraft outside of shielded areas, meaning that there is virtually no risk of collision with manned aircraft. Nonetheless, JSAs should ideally be performed prior to every operation, regardless of location to ensure familiarisation with the operating area and application of any appropriate risk mitigations. The need to consider the operating area and what to do if something goes wrong has been clearly established in academic literature [1,5]. Being unaware of the operating environment (both on the ground and in the air) is a common theme in unmanned aircraft incidents [44,46,70]. However, understanding how to identify, assess, and mitigate airborne and ground risks is something that is not innate and may require educational standards in place for unmanned aircraft users. Some of the risk mitigations also require training, for example, how to understand radio calls when using air band radio. This relates to the next area of discussion: why education is key.

### 4.2. Education Is Key

Those who did a course on unmanned aircraft were more likely to apply every risk mitigation than other users and were also more likely to apply a greater number than other users. This suggests that those who did a course had a more holistic view of risk mitigation for unmanned aircraft operations. Their answers also suggested a greater appreciation of when a particular risk mitigation might be appropriate. Specifically, while they were more likely to use air band radio, they were not more likely to use it in a shielded area or in a low fly zone. Shielded areas do not have manned aircraft within them, so there is no need to use air band radio in this environment. Equally, flying in low fly zones is something that is not permitted unless operating under a Part 102 Operator's certificate. Thus, it appears that those who had attended a course were able to identify that air band radio was not an appropriate risk mitigation in a shielded area and that it was not enough of a risk mitigation to operate within a low fly zone. Those who did a course were more confident in their ability to identify and comply with airspace requirements, which generally predicts accuracy. However, they were only more accurate than other users at using AirShare, not VNCs. This may suggest that AirShare is easier to learn or that current courses do not cover VNCs in sufficient detail for users to be able to apply them accurately. Given the similarity in AirShare and VNC accuracy amongst users overall, the latter appears to be more likely. Nonetheless, attending a course improves risk mitigation on almost every measure, suggesting that educational requirements may be an effective regulatory policy for mitigating the risk of unmanned aircraft operations. Taking a risk-based approach, this would not need to be for every user, but certainly would be helpful for operations where there are clear airborne and ground risks that need to be managed. This verifies past research that suggests pilot training is an effective risk mitigation in of itself [1,71].

### 4.3. OCAs

Those who completed OCAs employed more risk mitigations overall, and were also more likely to use MFNZ requirements, internal company procedures, or P102 procedures. These specific risk mitigations relate to OCAs because MFNZ requires OCAs for members to obtain their wings badge qualification, many companies require their staff to complete an OCA, and it is a requirement to pass an OCA to be authorised as a pilot for a Part 102 operation. Accordingly, these findings are to be expected. Those who completed OCAs also reported higher confidence levels for identifying and complying with airspace

requirements. However, these users did not actually perform any better than the rest of the sample with regard to AirShare and VNC accuracy.

### 4.4. User Type

In terms of the number of risk mitigations that are typically applied, the professional users employed more risk mitigations than semi-professional users, and both professional and semi-professional employed more than recreational users. Professional and/or semi-professional users were also more likely to use all of the specific risk mitigations (excluding pre-flight checks and MFNZ requirements). Both had higher confidence levels than recreational operators. Interestingly, professional operators were more accurate at reading VNCs than the rest of the sample, and semi-professional users were more accurate at using AirShare. VNCs do provide a far more comprehensive overview of airspace requirements, while AirShare focusses on specific airspace requirements that are more applicable to unmanned aircraft (e.g., 4 km boundaries around aerodromes). Perhaps this explains the difference in performance because the nuance involved with professional users' operations may necessitate the use of VNCs, while semi-professional users may get all the airspace information that they need from AirShare. Regardless of this area of difference, the use of a greater number and variety of risk mitigations than other users suggests that these users are distinct from recreational users. This is likely because their reputation and livelihoods depend on safe operations.

### 4.5. MFNZ and the Self-Regulation Model

MFNZ members employed less risk mitigations overall and were less likely to apply all of the specific risk mitigations, except for pre-flight checks and air band radio (where there were no differences with other users), and MFNZ site-specific requirements, which they were more likely to employ. MFNZ members were no more confident than other users at identifying and complying with airspace requirements, and they performed worse at AirShare accuracy (but not VNC accuracy). These findings show that MFNZ members are heavily reliant upon site-specific requirements to mitigate risk. This supports the idea of self-regulation for model aircraft clubs [49]. However, in New Zealand's case, this could be more formalised with the adoption of Part 149 to regulate member-based model flying organisations like MFNZ, in line with ICAO's model regulations [24]. This would then have the clear implication that any operations outside of what is allowed at their sites under Part 149 would come under the general Part 101 rules with the need to use other risk mitigations, such as using AirShare, checking VNCs and NOTAMs, and conducting JSAs. Because many of their sites are danger areas specifically designated for model aircraft flying, they can undertake more 'dangerous' operations, but often the same sorts of operations cannot be done safely outside of those.

### 4.6. Professional Bodies

UAVNZ members were more likely to employ a greater number of risk mitigations and were more likely to use every specific risk mitigation except for pre-flight checks and MFNZ requirements. There was no difference in their confidence levels against the rest of the sample; however, their members did perform worse at VNC accuracy than other participants. Thus, members of a professional body appear to take the need for risk mitigations more seriously, but that does not mean they necessary undertake those risk mitigations better than others (and in the case of VNCs, they appear to perform worse). Nonetheless, perhaps regulatory authorities should provide better recognition of professional bodies for unmanned aircraft users and collaborate with them to ensure that their members are not only employing more risk mitigations, but also that these are being employed accurately and appropriately. Professional bodies can also self-regulate by investigating potential breaches of their code of conduct by members and responding accordingly, potentially lessening the workload of regulators.

### 4.7. Part 102 Certification

Part 102 operators were more likely to employ a greater number of risk mitigations and were more likely to undertake all specific ones except for pre-flight checks and MFNZ requirements. They were more likely to use air band radio, especially for operations above 400 feet above ground level and in low fly zones. Part 102 operators also reported higher confidence levels in identifying and complying with airspace requirements but did not perform any differently to the rest of the sample at VNC or AirShare accuracy. The application of a greater number and variety of risk mitigations is not surprising as these form part of the safety case for Part 102 certification in the first place. In addition, Part 102 requires that the unmanned aircraft used are identifiable, meaning there is greater accountability if something did go wrong due to not applying sufficient risk mitigations. Thus, these findings validate the use of Part 102 to take a risk-based approach for approving variations to the Part 101 rules.

### 4.8. Limitations and Future Research

While this study helps to ameliorate a significant gap in the academic literature regarding understanding how unmanned aircraft users currently mitigate risk, it does so based upon a study conducted in a specific location. Due to the lack of international standardisation of unmanned aircraft regulations, and because each country has its own model flying organisations and professional bodies, the findings in this study may not be generalisable. To help better understand which findings translate and which are specific to New Zealand, future research could replicate this study in other countries.

This study was conducted as part of a wider research project examining unmanned aircraft users in New Zealand; thus, participants were also asked other questions about topics like unmanned aircraft ownership, types of operations conducted, effectiveness of safety promotion, and opinions about the current regulatory system. This study can therefore not rule out that answers may have been affected by participant fatigue or other such order effects [72]. Consequently, the replication of this study with only the specific areas of questioning related to risk mitigation would also help to verify the findings of this study.

Due to the use of convenience sampling, the sample may not be representative in terms of its proportionality of user types. Specifically, there were 650 MFNZ members out of a sample of 812. Because membership of MFNZ is not mutually exclusive with other demographic groupings, MFNZ members are potentially over-represented in general.

One of the more significant findings of this study is that education is key to improving risk mitigation; however, this was only measured by having attended a course on unmanned aircraft operations. Future research may like to examine the extent of informal education and its impact upon risk mitigation to see whether this also plays a role.

While this study captures the frequency of use of internal company procedures and Part 102 operating procedures, it does not probe further into the specifics of what these involve in terms of risk mitigations. Future research may wish to examine what companies and certified organisations are doing to mitigate operational risks.

## 5. Conclusions

This study presents the results obtained from a sample of 812 unmanned aircraft users regarding the number and variety of risk mitigations they typically apply, their confidence levels in identifying and complying with airspace requirements, and their accuracy in reading VNCs and using AirShare. The overall results show that the only widely used risk mitigation is performing a pre-flight check of the aircraft prior to an operation. This suggests that unmanned aircraft users need to view risk mitigation more holistically and move beyond only examining airworthiness. In particular, there are lower rates of using AirShare, checking VNCs and NOTAMs, performing JSAs, and listening to air band radio than what this study considers as ideal. It appears that the key way of increasing the number and variety of risk mitigations is through education, suggesting that educational

requirements may be a suitable policy direction. OCAs also yield improvements, but they are not as significant as attending courses. Professional and semi-professional operators, as well as those operating under Part 102 certificates, also utilise a greater number and variety of risk mitigations, presumably because their livelihoods depend upon safe operations and because there is greater reputational risk and greater potential for accountability than with other users. Members of MFNZ were heavily reliant upon MFNZ's site-specific requirements, suggesting New Zealand should investigate the use of Part 149 to better formalise the distinction between member-based model flying organisations and general operations under Part 101 rules. Members of the professional body UAVNZ did perform better in terms of the number and variety of risk mitigations, but not necessarily in terms of the accuracy at using them. Having members sign a code of conduct that is enforced by the organisation appears to work in improving risk mitigation; however, greater advice to members may be yielded through collaboration between professional bodies and regulatory authorities. In totality, the findings of this study contribute towards the academic literature by showing the prevalence of risk mitigations amongst unmanned aircraft users, as well as providing measures to increase the number and variety of risk mitigations in order to prevent incidents and accidents involving unmanned aircraft.

**Funding:** This research received no external funding.

**Institutional Review Board Statement:** This project was evaluated for ethical concerns by peer review and judged to be low risk. Consequently, it was not reviewed by one of Massey University's Human Ethics Committees but was registered as a low-risk project on the Massey University Human Ethics Database.

**Data Availability Statement:** The full list of survey questions (including questions not analysed in this article) and the data obtained from the wider study of unmanned aircraft users in New Zealand (of which the responses to the relevant questions were extracted for this article) can be accessed at the following link: https://doi.org/10.6084/m9.figshare.16571558.v1.

**Acknowledgments:** Thank you to Rick Watson, Andrew Shelley, Richard Milner, and Chris Jackson who piloted and provided useful suggestions to improve the survey prior to its launch.

**Conflicts of Interest:** Isaac Henderson is the Chair of UAVNZ, an industry and professional body representing commercial unmanned aerospace in New Zealand. He also consults for commercial unmanned aircraft operators domestically and internationally. However, none of the findings of this study benefit him directly or indirectly.

**Appendix A. Survey Questions**

The following questions were extracted from the wider dataset and the responses analysed for this article:

1. What is your gender?
    a. Male
    b. Female
    c. Other (please specify)
    d. Prefer not to say
2. What is your age?
3. Which of the following best describes you?
    a. Not a current unmanned aircraft user
    b. Recreational unmanned aircraft user
    c. Semi-professional commercial unmanned aircraft user (i.e., where less than 50% of your work time is spent on activities related to unmanned aircraft, including flight time, travel time, maintenance, data processing, etc.)
    d. Professional commercial unmanned aircraft user (i.e., where more than 50% of your work time is spent on activities related to unmanned aircraft, including flight time, travel time, maintenance, data processing, etc.)

4. Have you ever done a course on unmanned aircraft operations?

   a. Yes
   b. No

5. Have you ever passed an operational competency assessment (also known as a flight examination) on an unmanned aircraft?

   a. Yes
   b. No

6. Are you a member of Model Flying New Zealand?

   a. Yes
   b. No

7. Are you or your organisation a member of UAVNZ and/or Aviation New Zealand?

   a. Yes
   b. No

8. Have you ever operated under a Part 102 Operator's Certificate?

   a. Yes
   b. No

9. Which of the following do you typically do prior to an unmanned aircraft operation (select all that apply)?

   a. Log the flight on AirShare
   b. Check the visual navigation chart for the area
   c. Check for NOTAMs
   d. Conduct a job safety assessment
   e. Do a pre-flight check of your aircraft
   f. Other actions according to MFNZ site-specific requirements
   g. Other actions according to your company's internal procedures
   h. Other actions according to your company's approved Part 102 procedures

10. Do you ever use air band radio to monitor the local radio frequency when flying an unmanned aircraft?

    a. Yes
    b. No

11. [If participant answered "yes" to question 10] What sorts of operations do you typically use an air band radio for (select all that apply)?

    a. Operations within shielded areas
    b. Operations above 400 ft (120 metres)
    c. Operations within 4 km of an aerodrome
    d. Operations in low fly zones
    e. Operations in controlled airspace
    f. Other types of operations

12. How confident are you in your ability to identify different types of airspace and comply with their requirements (for example, restricted airspace or low fly zones)?

    a. Extremely confident
    b. Somewhat confident
    c. Neither confident nor unconfident
    d. Somewhat unconfident
    e. Extremely unconfident

13. Do you know how to read a visual navigation chart (VNC)?

    a. Yes
    b. No

14. [If participant answered "yes" to question 13] Looking at the following airspace from the VNC, which of the following is correct?

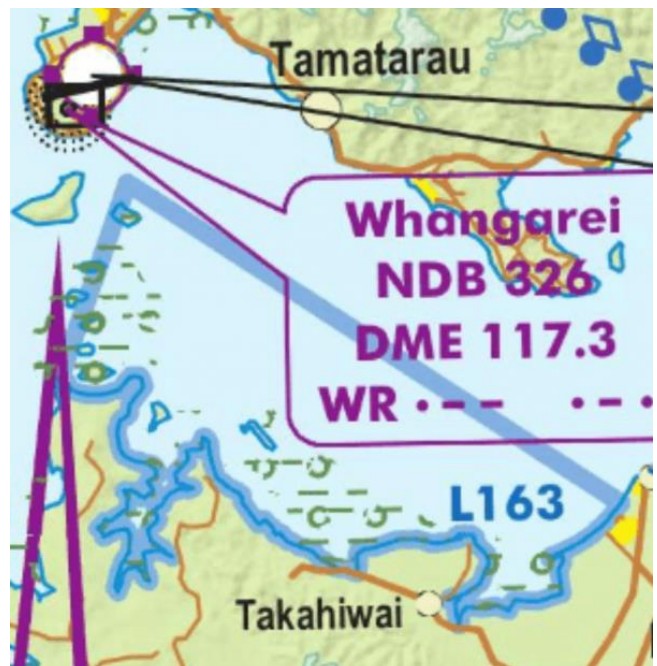

**Figure A1.** Excerpt from a VNC demonstrating a low fly zone (L163) that was presented to participants.

- a. General Aviation Area (GAA)
- b. Low Flying Zone L163 from SFC to 500 feet AGL **[correct answer]**
- c. Transit Lane 163
- d. Control Zone from SFC to 164ft AGL
- e. Unsure

15. [If participant answered "yes" to question 13] Looking at the following airspace from the VNC, which of the following is correct?

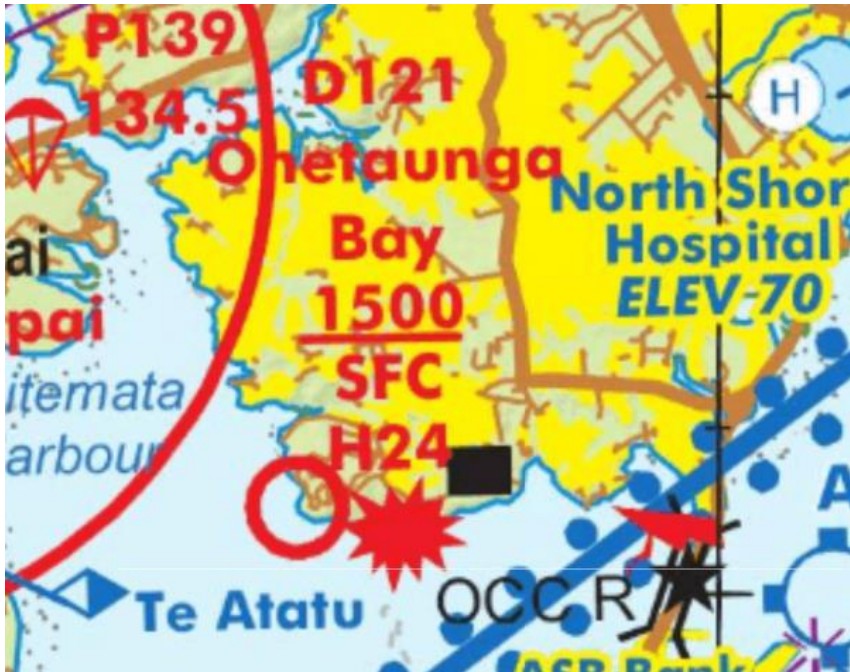

**Figure A2.** Excerpt from a VNC demonstrating a danger area for explosive hazards that was presented to participants.

    a.    Danger Area Ohetaunga. From SFC to 1500 m for Explosive Hazards. Active during the day.

    b.    Danger Area Ohetaunga. From 500 ft to 1500 ft for Explosive Hazards. Active during the day.

    c.    Danger Area Ohetaunga. From SFC to 1500 ft for Explosive Hazards. Active 24 h. **[correct answer]**

    d.    Danger Area Ohetaunga. From SFC to 1000 ft for Bombs. Active 24 h.

    e.    Unsure

16. Do you ever use AirShare to check airspace requirements?

    a.    Yes

    b.    No

17. [If participant answered "yes" to question 16] Using AirShare, you see that the area you wish to operate in is overlaid with a green area, such as in the picture below. What does this mean for your operation?

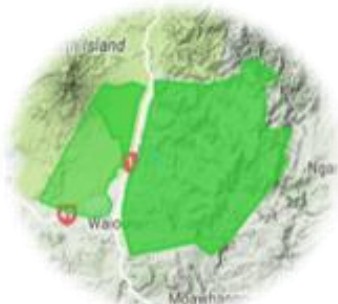

**Figure A3.** Excerpt from AirShare demonstrating a military operating area that was presented to participants.

    a.    There is a low fly zone—you cannot operate there under Part 101 rules

    b.    There is a control zone—you need to obtain permission from air traffic control

    c.    It is Department of Conservation land covered with restricted airspace—you need to obtain their permission as the administering authority

    d.    It is a military operating area—you need to obtain prior permission from the designated administering authority **[correct answer]**

    e.    Unsure

18. [If participant answered "yes" to question 16] Using AirShare, you see that the area you wish to operate in is overlaid with a red area, such as in the picture below. What does this mean for your operation?

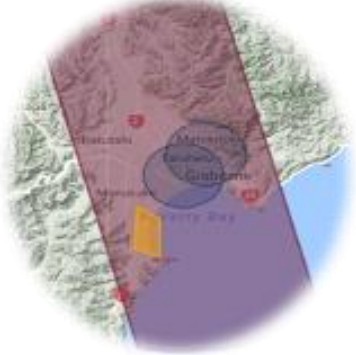

**Figure A4.** Excerpt from AirShare demonstrating a control zone that was presented to participants.

    a. There is a low fly zone—you cannot operate there under Part 101 rules

    b. There is a control zone—you need to obtain permission from air traffic control **[correct answer]**

    c. It is a military operating are—you need to obtain prior permission from the designated administering authority

    d. It is a danger zone—you must consider whether the danger in the area is a risk to your operation or vice versa

    e. Unsure

## Appendix B. Additional Statistical Reporting

*Appendix B.1. Associations between Specific Risk Mitigations and User Characteristics*

Table A1 presents the statistically significant results of the Chi-squared tests of independence to examine associations between user types and employing specific risk mitigations, with Cramer's V to measure the effect size of the association. The use of "Yes" or "No" in a column indicates the group that is more likely to use the risk mitigation (e.g., "Yes" under course means those who completed a course were more likely to apply the risk mitigation, while "No" under MFNZ would mean those who are not MFNZ members were more likely).

**Table A1.** Associations between specific risk mitigations and user characteristics.

| Risk Mitigation | User Type | Course | OCA | MFNZ | UAVNZ | Part 102 |
|---|---|---|---|---|---|---|
| Log flight on AirShare | Professional and Semi-Professional *** (V = 0.500) | Yes *** (V = 0.135) | - | No *** (V = 0.445) | Yes * (V = 0.073) | Yes *** (V = 0.216) |
| Read VNCs | Professional and Semi-Professional *** (V = 0.396) | Yes *** (V = 0.166) | No * (V = 0.069) | No *** (V = 0.341) | Yes *** (V = 0.144) | Yes *** (V = 0.212) |
| Check NOTAMs | Professional and Semi-Professional *** (V = 0.333) | Yes *** (V = 0.211) | - | No *** (V = 0.285) | Yes * (V = 0.070) | Yes *** (V = 0.252) |
| Conduct JSA | Professional and Semi-Professional *** (V = 0.503) | Yes *** (V = 0.208) | - | No *** (V = 0.267) | Yes ** (V = 0.092) | Yes * (V = 0.281) |
| Pre-flight check | - | Yes ** (V = 0.095) | - | - | - | - |
| MFNZ site-specific requirements | Recreational *** (V = 0.374) | Yes * (V = 0.070) | Yes *** (V = 0.371) | Yes *** (V = 0.648) | - | - |
| Internal company procedures | Professional *** (V = 0.635) | Yes *** (V = 0.256) | Yes ** (V = 0.114) | No *** (V = 0.265) | Yes *** (V = 0.133) | Yes *** (V = 0.369) |
| Part 102 exposition requirements | Professional *** (V = 0.613) | Yes *** (V = 0.204) | Yes *** (V = 0.146) | No *** (V = 0.153) | Yes *** (V = 0.202) | Yes *** (V = 0.599) |

Note: *, **, *** indicate statistical significance at the $p < 0.05$, 0.01, and 0.001 levels, respectively. Cramer's V = 0.1, 0.3, and 0.5 can be interpreted as small, medium, and large effect sizes, respectively [59].

*Appendix B.2. Number of Risk Mitigations Employed by Different Users*

A Kruskal–Wallis H test revealed statistically significant differences between the distribution of the number of risk mitigations for different user types, $\chi^2(2) = 128.948$, $p < 0.001$. Pairwise comparisons using Dunn's procedure [65] reveal significant differences in the mean rank number of risk mitigations employed between recreational users (372.64) and semi-professional users (603.42) ($p < 0.001$), recreational users (372.64) and professional users (727.66) ($p < 0.001$), and semi-professional users (603.42) and professional users

(727.66) ($p$ = 0.016). This establishes a clear hierarchy that professional users employ more risk mitigations prior to an operation than semi-professional users, and both employ more risk mitigations than recreational users.

Mann–Whitney U tests revealed the following findings regarding the distribution for the number of risk mitigations employed by each group:

1. The number of risk mitigations employed by those who completed a course (mean rank = 475.93) was statistically significantly higher than for those who did not (mean rank = 342.02), $U$ = 109462, $z$ = 8.758, $p$ < 0.001.
2. The number of risk mitigations employed by those who completed an OCA (mean rank = 428.54) was statistically significantly higher than for those who did not (mean rank = 351.72), $U$ = 80217.5, $z$ = 4.549, $p$ < 0.001.
3. The number of risk mitigations employed by MFNZ members (mean rank = 387.79) was statistically significantly lower than for non-members (mean rank = 481.56), $U$ = 40490, $z$ = −4.905, $p$ < 0.001.
4. The number of risk mitigations employed by UAVNZ members (mean rank = 456.24) was statistically significantly higher than for non-members (mean rank = 398.29), $U$ = 45798, $z$ = 2.645, $p$ = 0.008.
5. The number of risk mitigations employed by those who had operated under a Part 102 certificate before (mean rank = 600.45) was statistically significantly higher than for those who had not (mean rank = 380.80), $U$ = 52482.5, $z$ = 9.241, $p$ < 0.001.

*Appendix B.3. Associations between Use of Air Band Radio and User Characteristics*

Chi-squared tests of independence were run to see whether the use of air band radio in general and for specific types of operations was associated with particular user characteristics. The results of these tests are shown in Table A2 in the same manner as for Table A1.

**Table A2.** Associations between use of air band radio and user characteristics.

| Operation | User Type | Course | OCA | MFNZ | UAVNZ | Part 102 |
|---|---|---|---|---|---|---|
| Air band radio (in general) | Professional and Semi-Professional *** (V = 0.255) | Yes *** (V = 0.185) | Yes *** (V = 0.194) | - | Yes * (V = 0.083) | Yes *** (V = 0.301) |
| Shielded | - | - | - | No ** (V = 0.194) | - | - |
| Above 400 ft above ground level | Professional *** (V = 0.310) | Yes * (V = 0.177) | - | - | - | Yes *** (V = 0.281) |
| Within 4 km of an aerodrome | Professional and Semi-Professional ** (V = 0.219) | Yes ** (V = 0.204) | Yes *** (V = 0.312) | - | - | - |
| Low Fly Zone | Professional *** (V = 0.370) | - | - | No * (V = 0.171) | - | Yes *** (V = 0.258) |
| Controlled Airspace | Professional and Semi-Professional *** (V = 0.415) | Yes *** (V = 0.262) | Yes * (V = 0.153) | No *** (V = 0.259) | - | - |
| Other | Professional and Semi-Professional ** (V = 0.229) | - | - | No ** (V = 0.196) | - | - |

Note: *, **, *** indicate statistical significance at the $p$ < 0.05, 0.01, and 0.001 levels, respectively. Cramer's V = 0.1, 0.3, and 0.5 can be interpreted as small, medium, and large effect sizes, respectively [59]. For air band radio (in general) the sample size is 812, while for the types of operations the sample size is 207.

*Appendix B.4. Confidence Levels in Identifying and Complying with Airspace Requirements*

There were 808 participants who answered the Likert scale question regarding their confidence levels in their ability to identify different types of airspace and comply with their requirements. Of these participants, 417 (51.61%) indicated that they were "extremely confident", 304 (37.62%) indicated that they were "somewhat confident", 67 (8.29%) indicated that they were "neither confident nor unconfident", 15 (1.86%) indicated that they were "somewhat unconfident", and 5 (0.62%) indicated that they were "extremely unconfident". A single-sample *t*-test revealed a statistically significant difference between the mean score of 4.38 (*SD* = 0.766), which indicates somewhere between "somewhat confident" and "extremely confident", and the null hypothesis of 3 ("neither confident nor unconfident"), $t(807) = 51.097$, $p < 0.001$. Similarly, the Chi-squared goodness-of-fit test indicated that the proportion of participants in each category was statistically significantly different to the distribution expected by chance alone (i.e., 161.6 participants in each level of the scale), $\chi^2(4) = 869.252$, $p < 0.001$, mode = 5 ("extremely confident").

A Kruskal–Wallis H test revealed statistically significant differences between the distribution of confidence scores for different user types, $\chi^2(2) = 19.359$, $p < 0.001$. Pairwise comparisons using Dunn's procedure [65] revealed significant differences in the mean rank confidence scores between recreational users (392.59) and semi-professional users (487.67) ($p = 0.004$), and recreational users (392.59) and professional users (499.15) ($p = 0.004$). However, there were no statistically significant differences between the mean ranks of semi-professional users (487.67) and professional users (499.15). Thus, the interpretation is that recreational users are less confident in their ability to identify and comply with airspace requirements than semi-professional or professional users.

Mann–Whitney U tests revealed the following findings regarding the distribution of confidence scores for each group:

1.  The confidence scores of those who completed a course (mean rank = 429.04) were statistically significantly higher than for those who did not (mean rank = 381.83), $U = 91000$, $z = 3.194$, $p = 0.001$.
2.  The confidence scores of those who completed an OCA (mean rank = 415.74) were statistically significantly higher than for those who did not (mean rank = 376.76), $U = 73452$, $z = 2.392$, $p = 0.017$.
3.  The difference between the confidence scores for MFNZ members (mean rank = 398.11) and non-members (mean rank = 429.98) was not statistically significant, $U = 48197.5$, $z = -1.728$, $p = 0.084$.
4.  The difference between the confidence scores for UAVNZ members (mean rank = 427.18) and non-members (mean rank = 400.81) was not statistically significant, $U = 41830$, $z = 1.238$, $p = 0.216$.
5.  The confidence scores for those who had operated under a Part 102 certificate before (mean rank = 451.89) were statistically significantly higher than for those who had not (mean rank = 398.19), $U = 38370$, $z = 2.343$, $p = 0.019$.

*Appendix B.5. VNC Accuracy*

There were 804 participants who answered whether they know how to read VNCs or not. Of these, 325 (40.42%) indicated that they know how, while 479 (59.58%) indicated that they did not know how. Out of the 325 who said they knew how to read a VNC, 282 completed the two follow-up questions to assess their level of accuracy in identifying airspace requirements using VNCs. Of these 282 participants, 162 (57.45%) got both questions correct, 72 (25.53%) got one correct, and 48 (17.02%) got both questions wrong. More got the second question regarding the danger area correct (221 participants, 78.37%) than those that correctly identified the low fly zone in the first question (175 participants, 62.06%).

A Kruskal–Wallis H test revealed statistically significant differences between the distribution of VNC scores for different user types, $\chi^2(2) = 11.990$, $p = 0.002$. Pairwise comparisons using Dunn's procedure [65] revealed a significant difference in the mean rank VNC scores between recreational users (134.46) and professional users (182.64) ($p = 0.003$).

However, there were no statistically significant differences in the mean rank VNC scores between recreational users (134.46) and semi-professional users (153.15), nor between semi-professional users (153.15) and professional users (182.64).

Mann–Whitney U tests revealed the following findings regarding the distribution for VNC scores for each group:

1. The difference in VNC scores between those who had attended a course (mean rank = 144.27) and those who had not (mean rank = 138.07) was not statistically significant, $U = 10260$, $z = 0.714$, $p = 0.475$.
2. The difference in VNC scores between those who completed an OCA (mean rank = 144.64) and those who did not (mean rank = 130.81) was not statistically significant, $U = 7660$, $z = 1.343$, $p = 0.179$.
3. The difference in VNC scores between MFNZ members (mean rank = 139.72) and non-members (147.10) was not statistically significant, $U = 6895$, $z = -0.732$, $p = 0.464$.
4. The VNC scores for UAVNZ members (mean rank = 118.88) were statistically significantly lower than for non-members (mean rank = 146.61), $U = 4804$, $z = -2.493$, $p = 0.013$.
5. The difference in VNC scores between those who had operated under a Part 102 Operator's certificate before (mean rank = 153.09) and those who had not (mean rank = 138.94) was not statistically significant, $U = 6481$, $z = 1.262$, $p = 0.207$.

A cumulative odds ordinal logistic regression revealed that an increase in confidence levels (expressed as a score of 1–5 on a Likert scale) was associated with an increase in the odds of getting both VNC questions correct, with an odds ratio of 1.558, 95% CI [1.054, 2.304], Wald $\chi^2(1) = 4.935$, $p = 0.026$.

*Appendix B.6. AirShare Accuracy*

There were 761 participants who answered whether they ever use AirShare to check airspace requirements. Of these participants, 223 (29.30%) indicated that they do, while 538 (70.70%) indicated that they do not. Out of the 223 participants that used AirShare to check airspace requirements, 213 answered the two follow-up questions to measure their accuracy when doing so. From these 213 participants, 122 (57.28%) got both questions correct, 56 (26.29%) got one correct, and 35 (16.43%) got both wrong. More got the second question regarding the control zone correct (171 participants, 80.28%) than those who correctly identified the military operating area in the first question (129 participants, 60.56%).

A Kruskal–Wallis H test revealed statistically significant differences between the distribution of AirShare scores for different user types, $\chi^2(2) = 17.461$, $p < 0.001$. Pairwise comparisons using Dunn's procedure [65] revealed a significant difference in the mean rank AirShare scores between recreational users (95.48) and semi-professional users (133.22) ($p < 0.001$). However, there were no statistically significant differences in the mean rank AirShare scores between recreational users (95.48) and professional users (118.14), nor between semi-professional users (133.22) and professional users (118.14).

Mann–Whitney U tests revealed the following findings regarding the distribution for AirShare scores for each group:

1. The AirShare scores of those who completed a course (mean rank = 119.04) were statistically significantly higher than for those who did not (mean rank = 93.65), $U = 7004$, $z = 3.378$, $p < 0.001$.
2. The difference in AirShare scores between those who completed an OCA (mean rank = 108.89) and those who did not (mean rank = 104.15) was not statistically significant, $U = 5682.5$, $z = 0.620$, $p = 0.536$.
3. The AirShare scores for MFNZ members (mean rank = 99.36) were statistically significantly lower than for non-members (mean rank = 115.31), $U = 4813.5$, $z = -2.123$, $p = 0.034$.

4.  The difference in AirShare scores between UAVNZ members (mean rank = 113.24) non-members (mean rank = 105.86) was not statistically significant, $U$ = 3176, $z$ = 0.712, $p$ = 0.476.

5.  The difference in AirShare scores between those who had operated under a Part 102 Operator's certificate before (mean rank = 117.59) and those who had not (mean rank = 104.08) was not statistically significant, $U$ = 4328, $z$ = 1.481, $p$ = 0.139.

A cumulative odds ordinal logistic regression revealed that an increase in confidence levels (expressed as a score of 1–5 on a Likert scale) was associated with an increase in the odds of getting both AirShare questions correct, with an odds ratio of 1.622, 95% CI [1.034, 2.543], Wald $\chi^2(1)$ = 4.437, $p$ = 0.035.

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
