# Peer review of "Examining New Zealand Unmanned Aircraft Users’ Measures for Mitigating Operational Risks"

_drones, doi:10.3390/drones6020032_

Round 1
Reviewer 1 Report
Nice paper, in general I liked it quite much. I have some comments on some details I think should be clarified or modified to further enhance it:
- Table 1 risk mitigations are a little confusing, as here we have different things identified as mitigations that are not. For instance: Procedures, Operate in danger areas, Operate in controlled aerospace, Flight planning, organizational procedures, night operations, aircraft size/weight, … and many others are not risk mitigations but operating conditions, or they may cover not only safety related aspects but others (organizational, administrative, economic, …). I would try to improve the table by describing things potentially impacting safety from actual risk mitigations.
- In table 1 risk mitigations for airborne and ground risks are quite repetitive. Maybe reorganizing the table as Ground/Airborne, Airborne, Ground, would maje clearer which are the specific risk mitigations for each kind of risk.
- Line 82: there is an “s” for “is”, I think.
- Line 191: “fool proof” sounds too conversational. I would rewrite it to something as “free of issues”
- Line 192: There is another “s”, here I think instead on “a”
- Section 1.4.5 starts with the description of UAVNZ members. I think UAVNZ must be presented in the paper somehow, maybe just bellow the description of MFNZ in section 1.3.6. in a new section, or just at the beginning of section 1.4.5.
- Lines 262-263 are referring to ethical concerns. I think there is a proper section in the paper template to write about this. In any case, the lower risk here is confusing, being the paper about safety risks. So, if you maintain this sentence, please put it more in context.
- The potential bias of the study due to the high prevalence of MFNZ members, and the much lower representation of other types of drone users, should be emphasized in the discussion and conclusions. Some details regarding the posibility that a drone user belongs to more than one class should also be discussed (i.e. a drone user at the same time part of MFNZ, UAVNZ, and using Part 102, …
- Lines 334-337 have repeated information, please rephrase it.
- A general problem in the paper, regarding results, is the lack of graphical representations. Quite often many of the data in tables could be summarized in figures easying its comprehension.
- Also, the complementarity/potential redundancy of some of the information (VNC, Airshare, Notams), should be discussed, and maybe aggregated statistics should be included (i.e. users consulting aerospace information, users consulting real time information, …)
- Similarly with the presence of formal/informal education.
- Finally, some of the mitigation methods are unclear, such as Internal company procedures, MFNZ site specific requirements, or Part 102. Could you provide further examples/insight on the basis of your survey responses?
Author Response
Thank you for your time taken to review this manuscript. The comments were helpful in improving the paper. Attached is a document where each comment is responded to.

Reviewer 2 Report
Strengths:
- Addressing an interesting gap in the literature
- Publicly accessible survey results that can spark further interest in this area
Weaknesses:
- Limited analysis and consideration of correlated quantities (e.g., users belonging to multiple groups, overlap in the risks that are mitigated by different approaches)
Overall summary:
The paper presents an interesting analysis on how users implement risk citation strategies when performing UAS operations. This work tries to bridge the gap between several academic studies on risk mitigation strategies with what is practical and already being implemented by users. Although the scope of the study is limited to the New Zealand context, I believe that it is an important first step in addressing a broader question on understand compliance amongst UAS operators. However, I do have some concerns with the manuscript in its current form.
Main comments:
- The author presents several risk mitigation strategies to be chosen in a survey. It seems like many of the strategies presented are not mutually exclusive. For example, Part 102 procedures set by a company might require checking for NOTAMS. However, I do not see an analysis of this effect.
- The author states initially that they want to study voluntary risk mitigation strategies. However, I am not sure that all the options presented are voluntary. Does compiling with a legal requirement for 102 operations count as voluntary compliance? This aspect needs to be clarified in the paper.
- Is it possible that the site-specific requirements covers a lot of the other risk mitigation measures? If so, that explains why the MFNZ members do not seem to follow other measures.
Minor comments:
- Users being parts of multiple groups may also explain some of the results. Has that possibility been considered?
- The manuscript is longer than necessary. Many of the analysis can either be moved to a table, the appendix, or a figure. For example, the Mann-Whitney U test results could be presented in a shorter way with details moved to the appendix.
- Section 1.3.6: “operations are done in danger areas that are designated…” Maybe the authors can clarify what they mean by a “danger area” — I presume its an area that is dangerous for manned aircraft to fly (and not a danger area where drones should not fly)?
- I am not sure why the number of risk mitigation strategies is a valuable measure. From a high-level, I would assume that the effectiveness of the strategy, the nature of risk that it mitigates, and the degree of non-overlap in the risks mitigated will play an equally important role. Can the author elaborate more on their reasoning?
- Doesn’t all airborne risk mitigation also result in reduced ground risk (by mitigating the danger of falling debris)? Maybe worth clarifying the distinction between airborne and ground risk a little more clearly.
- I think a better organization might be to give a preview of the main hypothesis that are going to be tested before the results. Next, present a quick summary of the statistical tests that will be used to investigate these hypothesis. Finally, present the results (with the more involved details that are not required for understanding the key takeaways in the appendix).
Editorial comments:
- Abstract: Can shorten to “manned aircraft users in New Zealand, their confidence levels …”
- Abstract: Can shorted to “the number and type of risk mitigation strategies applied, users’ preference….”
- Section 1.4: Please use “this study illustrates/demonstrates/finds” instead of “this study predicts”
I would hope that a revised version of the manuscript improves the analysis of the potentially overlapping and correlated data fields, and addresses most of my minor comments (which are relatively easy to incorporate).
Author Response
Thank you for your valuable time given to reviewing this manuscript. The comments were helpful towards improving the paper. Each comment has been responded to in the attached document.

Reviewer 3 Report
I congratulate the author because the information in this article clearly illustrates the issues, problems, and trends related to the topic of unmanned aircraft users measures for mitigating operational risks in New Zealand. The proper emphasis is given to fulfil the hypothesis of this article, which is to help to ameliorate the gap in the literature related to the unmanned aircraft users measures for mitigating operational risks in New Zealand, by examining how unmanned aircraft users in New Zealand currently mitigate their operational risks. The article is clearly scientific, and the terminology is adequate.
It is plausible the decision to focus on eight pre-flight risk mitigations and the use of air band radio as ways that users could reduce ground and airborne risks associated with their operation. The emphasis dedicated to these eight pre-flight risks mitigations and their analysis is quite accurate. It is very clear why has been chosen New Zealand as a study location and the materials and methods used for this research.
I kindly ask the author to add a reference to the international background related to the unmanned aircraft measures for mitigation operational risks before explaining New Zealand as a study site.
I strongly support the publication of this article and once more congratulations to the author for the effort and dedication.
Author Response
Thank you for your valuable time spent reviewing this manuscript. The specific comments have been responded to in the attached document.
